# The Response of Prostate Cancer to Androgen Deprivation and Irradiation Due to Immune Modulation

**DOI:** 10.3390/cancers11010020

**Published:** 2018-12-26

**Authors:** Chun-Te Wu, Wen-Cheng Chen, Miao-Fen Chen

**Affiliations:** 1Department of Urology, Chang Gung Memorial Hospital at KeeLung, KeeLung 20401, Taiwan; wucgmh@gmail.com; 2School of Medicine, College of Medicine, Chang Gung University, Taoyuan 33302, Taiwan; rto_chen@yahoo.com.tw; 3Department of Radiation Oncology, Chang Gung Memorial Hospital at Chiayi, Chiayi 61363, Taiwan

**Keywords:** irradiation, abscopal effect, ADT, immune, prostate cancer

## Abstract

This study investigated changes in the immune system and the biological consequences of androgen deprivation therapy (ADT) and radiotherapy (RT) for augmenting the treatment response in prostate cancer, particularly for castration-resistant prostate cancer (CRPC). Human and murine prostate cancer cell lines were used to examine the response to ADT and RT in vitro and in vivo. Biological changes following treatment and related immune modulation in the tumor microenvironment were examined. Our results showed that CRPC cells were demonstrated to be more resistant to the RT and ADT treatments. ADT increased tumor inhibition following irradiation. The underlying changes included increased cell death, attenuated myeloid-derived suppressor cell recruitment, and an increase in the number of tumor-infiltrating T cells (TILs). Furthermore, when high-dose fractionated RT was given to the primary CRPC tumor, a smaller size of secondary non-irradiated tumor associated with increased TILs was noted in ADT-treated mice. In conclusion, treatment resistance in CRPC was associated with a more immunosuppressive microenvironment. Enhanced antitumor immunity was responsible for the augmented RT-induced tumoricidal effect induced by ADT. Immune modulation could be a promising strategy for prostate cancer, especially for metastatic CRPC.

## 1. Introduction

Although most prostate cancers show an initial favorable regression following androgen deprivation therapy (ADT), the development of resistance is inevitable, and the resulting form of the disease is called castration-resistant prostate cancer (CRPC) [1,2]. In the clinic, CRPC is usually highly aggressive and responds poorly to treatment. Radiotherapy (RT) is the standard of care for localized prostate cancer [3]. The response of tumors to RT is multifactorial and depends on the intrinsic sensitivity of the tumor cells, as well as features of the tumor microenvironment, including immune responses [4]. This immune modulation following RT includes altered cytokine signaling and the recruitment of immune regulatory cells. We have previously reported that inflammation associated with the activation of myeloid-derived suppressor cells (MDSCs) and an increase in regulatory T cells is critical for aggressive tumor behavior and the poor radiation response [5,6]. Much of the current interest in combining immunomodulation and RT lies in strategies to overcome the persistent suppression of adaptive immune responses by the tumor and its microenvironment [7,8]. However, the appropriate treatment scheme for prostate cancer remains unclear.

Adding ADT to RT appears to improve the outcome by enhancing both local and distant disease control. For many years, ADT and RT were presumed to work through a direct cytotoxic action on tumors: However, recent studies have uncovered under-appreciated benefits of these treatments for the immune system. ADT and RT modulate the immunity of a tumor by regulating both local and systemic molecular and cellular responses [9]. A number of possible mechanisms have been identified by which ADT and RT improve disease control [10,11], but the synergistic mechanisms are still unclear. Accordingly, combined treatment (ADT and RT) may require further immunological investigation for a full conversion to effector activity. This study investigated changes in the immune system and the biological consequences of combined therapy to aid in the development of new strategies for augmenting the treatment response of prostate cancer or CRPC.

## 2. Materials and Methods

### 2.1. Cell Culture and Reagents

Human LNCaP, an androgen-sensitive prostate cancer cell line, was obtained from the American Type Culture Collection (Manassas, VA, USA), and maintained in Roswell Park Memorial Institute (RPMI) medium (Gibco, Carlsbad, CA, USA) with 10% fetal bovine serum (FBS). The respective CRPC cells, LNCaP-hormone resistant (HR) cells [12], were obtained from LNCaP cells after long-term (>16 weeks) culture in RPMI with 10% FBS and 2 mM bicalutamide (Astra-Zeneca, Cambridge, UK). A transgenic adenocarcinoma of the mouse prostate cancer cell line (TRAMP)-C1 was kindly gifted from Dr. Ji-Hong Hong [5,13]. TRAMP-C1 cells were cultured in Dulbecco’s Modified Eagle’s Medium (DMEM) supplemented with 10 nM dehydroisoandrosterone and 10% FBS. To establish the CRPC cells, we cultured TRAMP-C1 in androgen-deprived medium (DMEM with 10 nM flutamide and 10% dextran-coated charcoal-treated FBS). After >12 weeks of culture, these cells grew significantly faster than TRAMP-C1 in androgen-containing medium and were designated TRAMP-HR [5]. The cells were treated with vehicle or 10 μM of enzalutamide for in vitro ADT.

### 2.2. Clonogenic Assay

To determine the intrinsic cellular radiosensitivity, we used a clonogenic assay. Exponentially growing cells were irradiated with single doses of 0, 2, 4, or 6 Gy using a 6-MeV electron beam, and then immediately counted, diluted, and plated onto 60-mm culture dishes. After incubation at 37 °C for 10 days, the plates were stained with crystal violet (Sigma) for colony counting. Colonies containing more than 50 cells were scored, and plating efficiency and surviving fractions were determined for each cell line. The survival fractions were determined by measurement colony after irradiation and divided by plating efficiency. To determine the effects of concurrent ADT treatment on radiation-induced cell death, cells were pretreated with 10 μM enzalutamide before irradiation. After irradiation, enzalutamide was retained in the cell culture.

### 2.3. Ectopic and Orthotopic Mouse Tumor Models and Radiation

Eight-week-old male C57BL/6J and Balb/c nude mice were used as the tumor implantation model, with the approval of the experimental animal committee of our hospital. (The approval code for the study is 2017102601 from Institutional Animal Care and Use Committee of Chang Gung Memorial Hospital, and the date I got the approval is 27 November 2017). In the ectopic tumor implantation model, hormone-sensitive (HS) and the respective CRPC cells (1 × 10^6^ cells in 30 μL PBS per implantation, six animals per group at least) were subcutaneously implanted into the dorsal gluteal region. In the orthotopic tumor implantation model, prostate cancer cells (6 × 10^6^ cells in 50 μL PBS per implantation, six animals per group at least) were intraoperatively implanted into the lateral region of the prostate gland [5]. The extent of orthotopic tumor invasion and tumor size were measured 2 weeks after implantation or at the indicated times, as we previously reported [14]. To determine radiosensitivity in vivo, local 15 Gy irradiation was performed when ectopic tumors reached 0.5 cm^3^, or two weeks after implanting the orthotopic tumor. Control mice were subjected to sham irradiation. The growth curves in mice exposed to irradiation were determined by the relative tumor volume normalized to the tumor size at the time of irradiation. The radiosensitivities of different xenografts were indicated by growth delay (i.e., the time required after irradiation for the tumor to recover its previous volume). To investigate the abscopal effect on tumor regression [15], the cells were simultaneously injected into the right thigh (primary tumor) and left upper back (secondary tumor). Hypofractionated regimens such as 3 × 8 Gy are reported to be more efficient with respect to the abscopal response of radiation than single high/ablative doses in vivo [16,17]. Therefore, the primary tumor was irradiated three times with 8 Gy in one week, and tumor sizes (including primary irradiated and secondary non-irradiated tumors) were measured at the indicated times thereafter. To evaluate the effect of ADT, a bilateral orchiectomy was performed one week before tumor implantation in the mice assigned to the surgical ADT group. An intraperitoneal (i.p.) injection of 25 mg/kg enzalutamide was given daily to the medical ADT group [18]. To test the effect of ADT on the radiation response in vivo, we performed surgical orchiectomy or started enzalutamide (25 mg/kg enzalutamide daily) treatment one week before the start of RT. The effects of inhibiting interleukin (IL)-6 on tumor growth were also investigated in vivo. The treatment regimen consisted of a weekly i.p. injection of anti-IL-6 or isotype antibody at 0.5 mg/mouse.

### 2.4. Myeloid-Derived Suppressor Cells (MDSCs) and CD8+ T Cells for Flow Cytometric Analyses

MDSCs are characterized by co-expression of the myeloid cell lineage differentiation antigens Gr1 and CD11b [19]. Therefore, we used the specific Gr1 antibody (Clone RB6-BC5), which reacts with a common epitope on Ly-6G and Ly-6C, and the antibody specific to CD11b (Clone M1/70) (BD Pharmingen, San Jose, CA, USA), to define mouse MDSCs as CD11b+Gr1+ in this study. In addition, we used antibodies specific for CD3 and CD8 to define tumor-infiltrating T cells (TILs) and cytotoxic T cells in murine tumors, respectively. For fluorescence-activated cell sorting (FACS) analysis, the tissue specimens (three mice per group, duplicates) were cut into pieces and further digested to isolated cells in RPMI 1640 medium containing 0.05 mg/mL liberase and 0.1 mg/mL DNAse in an incubator at 37 °C for 40 min [20]. FACS analysis was carried out on single-cell suspensions prepared from whole tumors after digestion and immunostaining for CD3 or CD8 with fluorescence-labeled monoclonal antibodies (BD Pharmingen). The percentage of MDSCs and T cells was measured by multicolor flow cytometry with the abovementioned monoclonal antibodies. Isotype-specific antibodies were used as negative controls in FACS.

### 2.5. Immunohistochemical (IHC) Staining and Immunofluorescence (IF) of Tissue Specimens

Formalin-fixed, paraffin-embedded tissues were cut into 5-μm sections for IHC analysis. Antibodies specific for IL-6, p-H2AX, Ki-67, and active caspase 3 were obtained from Santa Cruz Biotechnology, Inc. (Santa Cruz, CA, USA), Research & Diagnostics Systems, Inc. (Minneapolis, MN, USA), Cell Signaling Technology (Danvers, MA, USA), and Chemicon (Temecula, CA, USA). The sections were incubated overnight at 4 °C with antibodies against the target proteins. Frozen tissue specimens were cut into 5–8 μm cryostat sections, warmed to room temperature, fixed for 10 min in cold acetone (−20 °C), and incubated for 20 min in PBS containing 10% goat serum. The sections were incubated overnight at 4 °C with antibodies against IL-6, CD11b, and CD3, washed three times with PBS, and incubated for 1 h with fluorescein or Texas Red-conjugated secondary antibodies. The positive staining signals were assessed by microscope from ten random fields and semiquantitated by MetaMorph software (version 7.7, Molecular Devices, Sunnyvale, CA, USA)

### 2.6. Statistical Analysis

The significance of the differences between the samples was determined using Student’s *t*-tests. The data are presented as the mean ± standard error of the mean. All experiments, comprising three replicates at least, were performed twice or thrice independently. A probability level of *p* < 0.05 was considered statistically significant, unless otherwise stated.

## 3. Results

### 3.1. Role of ADT in the Response to RT

By cellular experiments and ectopic tumor growth, ADT was shown to significantly inhibit tumor growth through increased cancer cell death (Figure 1). Furthermore, the effects of ADT on radiation sensitivity were determined by clonogenic assay and delay in tumor growth. In vitro, the human LNCaP and murine TRAMP-C1 prostate cancer cell lines were exposed to single radiation doses of 0, 2, 4, or 6 Gy in the presence or absence of ADT, and their survival curves were determined by colony-forming assays. As shown in Figure 1a,b, ADT increased the RT-induced loss of clonogenic cells, which was associated with increased cell death. Due to the persistence of p-H2AX linked to the formation of DNA double-strand breaks and response to RT, we examined p-H2AX in situ by IF. Figure 1c demonstrates that ADT increased the DNA damage 24 h after RT. The analysis of ectopic tumors in immunocompromised mice (Figure 1d) confirmed the in vitro findings, to show that ADT increased the response of prostate cancer to radiation.

### 3.2. Response to Radiation Treatment in CRPC

In vitro, HS and CRPC Cells were exposed to single radiation doses of 0, 2, 4, or 6 Gy in the presence or absence of ADT, and their survival curves were determined by colony-forming assays. The data revealed that CRPC cells had more resistance to RT and the ADT treatment compared to the respective HS cells (Figure 2a). Moreover, as shown in Figure 2b,c, the CRPC cells exhibited less cell death and DNA damage induced by RT or ADT. The CRPC cells had larger tumor xenografts following RT and ADT treatments of immunocompromised mice (Figure 2d). Moreover, ADT attenuated the RT resistance of CRPC, as demonstrated by the smaller tumor sizes associated with increased RT-induced cell death compared to RT alone (Figure 2a–d).

### 3.3. Radiation Response of Prostate Cancer in Immunocompetent Mice

To investigate the role of the immune tumor microenvironment in the radiation response of prostate cancer, we examined the radiation response of prostate cancer using ectopic and orthotopic tumor animal models in immunocompetent mice. Figure 3a,b shows that CRPC tumors (TRAMP-HR) were more resistant to RT, as demonstrated by a shorter duration tumor growth delay following irradiation of subcutaneous tumors, and larger orthotopic tumors compared to HS tumors. Moreover, Figure 3c shows that the RT-induced DNA damage was significantly higher in HS tumors, associated with attenuated staining of the cellular proliferation marker Ki-67, compared to that noted in HR tumors. It was assumed that MDSC recruitment played a role in tumor regrowth after irradiation [21]. Accordingly, recruitment of MDSCs was examined in tumor-bearing mice following irradiation. FACS and immunofluorescence analyses demonstrated that irradiation stimulated MDSC recruitment (Figure 3d–f). Moreover, the levels of MDSCs in HR tumors were significantly greater than those implanted with HS tumor cells, with or without RT. We previously reported a positive link between the IL-6 level and MDSC recruitment for prostate cancer [6]. Therefore, we further examined IL-6 levels in tumors and the sera of tumor-bearing mice. As shown in Figure 3f,g, HR cells had higher IL-6 expression, and irradiation augmented the increased IL-6 secretion.

### 3.4. Role of ADT in the Radiation Sensitivity of CRPC in Immunocompetent Mice

Because ADT has been reported to possess an immune modulatory effect on prostate cancer [22], we further examined if the immune modulatory effects of ADT play a role in the radiation sensitivity of CRPC in immunocompetent mice. As demonstrated in Figure 4a–b, ADT prolonged the RT-induced tumor growth delay in the ectopic tumor model, and smaller tumors were observed in the irradiated orthotopic tumor model of immunocompetent hosts. The immunohistochemistry analysis using orthotopic tumors (Figure 4c) and FACS analysis using murine spleens (Figure 4d) showed that ADT increased RT-induced DNA damage, which was associated with attenuated MDSC recruitment after irradiation. Infiltration of T cells into tumors is correlated with improved prognosis in several types of cancers [23]. Furthermore, the presence of MDSCs inhibits antitumor T-cell responses [20]. The data in Figure 4e show that ADT increased the infiltration of T cells in irradiated tumors. These data suggest that the attenuation of the RT-induced MDSC recruitment and the increased TILs in tumors were associated with the radiosensitization induced by ADT.

### 3.5. Role of High-Dose RT in the Response of CRPC to ADT

High-dose RT has the ability to alter the immunosuppressive tumor environment and induce an immune-mediated abscopal effect [18,24]. To investigate if high-dose RT augmented the tumor inhibition of metastatic CRPC in mice treated with ADT, we simultaneously implanted CRPC tumor cells into the upper back (secondary non-irradiated tumor) and right thigh of the same mice subcutaneously, and irradiated the right thigh tumor only (primary irradiated tumor). Micro-positron emission tomography images were taken 24 h after irradiation, the mice were analyzed, and their tumors were removed for further evaluation 1 week after local RT. As shown in Figure 5a–c, smaller tumors tended to be associated with a lower standardized uptake value ratio, and decreased Ki-67 was noted in the secondary non-irradiated tumors of the ADT + RT group compared to those in the ADT alone group. In addition, RT to the primary tumor was linked to increased numbers of TILs in secondary non-irradiated tumors of mice that underwent ADT (Figure 5d). Based on these data, high-dose local RT may augment the response to ADT in non-irradiated tumors by increasing antitumor immunity.

### 3.6. Inhibiting IL-6 Enhanced the RT-Induced Abscopal Effect on CRPC

IL-6 plays an important role in the induction of MDSC recruitment, and subsequently contributes to increased resistance following irradiation [25]. As shown in Figure 6a–b, inhibiting IL-6 augmented the RT-induced tumoricidal effect in irradiated tumors, and induced greater regression of non-irradiated tumors in mice treated with ADT. Inhibition of IL-6 attenuated MDSC recruitment in the tumors and spleens of tumor-bearing mice (Figure 6c), increased CD3+ TILs (Figure 6d), and increased CD8+ cytotoxic cells (Figure 6e) in non-irradiated tumors of mice that underwent ADT or local RT. These results suggest that IL-6 inhibition increased the antitumor immune response and might augment the abscopal effect induced by high-dose local RT.

## 4. Discussion

The combination of ADT and RT has been shown to improve overall and prostate cancer-specific survival over RT alone in patients with prostate cancer. We showed that the combination of RT with ADT induced more cell death in vitro and a longer tumor growth delay in vivo than in prostate cancer treated with RT alone. In the present study, the ADT treatment included surgical orchiectomy and the androgen receptor (AR) antagonist enzalutamide [26]. A number of possible mechanisms by which ADT and RT improve disease control have been proposed, but the synergistic mechanisms are poorly understood. The ability of ADT to initially reduce the tumor burden makes it a cornerstone of prostate cancer treatment. Most patients, however, eventually develop CRPC that progresses rapidly despite ongoing systemic hormone suppression. The realization that CRPC remains fueled by androgen signaling, albeit with an increased utility of nontraditional pathways and alterations involving the androgenic ligand, has led to newer AR-directed agents [2,27,28]. In the clinic, CRPC usually presents as highly aggressive and responds poorly to treatment. We further used CRPC cancer cells to examine the effect of ADT on the radiation response in vitro and in tumor-bearing nude mice. The data revealed that CRPC had a poor response to RT and ADT, and was associated with decreased RT-induced cell death and DNA damage compared to that of HS cells.

A systemic immune response is required for tumor rejection. There is a general consensus that TILs play a role in the recognition and elimination of tumor cells, and are associated with better patient outcomes [29,30]. A tumor’s “immunological status” is of great importance when considering how to best enhance the response to treatment [31]. In addition to directly damaging DNA, the radiation-induced response is dynamic and involves several mediators and immune responses [32,33]. To study the relationship between the immune reaction and the radiation response, we examined the radiation response of prostate cancer in immunocompetent mice. The data revealed that CRPC with a poor RT response was associated with decreased DNA damage, higher IL-6 levels, increased recruitment of MDSCs, and attenuated TILs. These findings suggest that induction of a more immunosuppressive tumor environment plays a critical role in the poor radiation response of CRPC.

New data has highlighted the importance of AR signaling in immune regulation. Androgens regulate a variety of immune responses, and the effects of ADT on the immune system have been previously reported [34]. It appears that the androgen–AR axis has a profound suppressive effect on the behavior of various lymphocyte subsets. Accordingly, we further examined the role of ADT in the radiation response of CRPC and its relationship with immune regulation. There was a longer RT-induced tumor growth delay associated with increased DNA damage in ADT-treated mice compared to those without ADT. Moreover, FACS and immunofluorescence analyses revealed that there were more TILs associated with attenuated MDSC recruitment in CRPC tumors following RT in ADT-treated mice. In addition to surgical ADT (orchiectomy), enzalutamide currently plays a major role in the management of recurrent prostate cancer, or CRPC. AR antagonists may have different effects on immune modulation than depletion of testosterone [22]. We found that the tumor growth delay associated with increased TILs and attenuated MDSC recruitment induced by enzalutamide were similar to that induced by castration. Based on these data, we suggest that an immune-mediated mechanism could explain the radiation sensitization effect induced by ADT (including surgical and medical), at least in part.

RT has the ability to induce an immune-mediated abscopal effect [15,24]. This effect is associated with proimmunogenic effects prevailing over immunosuppressive effects. Preclinical models have established that the abscopal effect is T cell-dependent [35]. Radiation-dosing regimens identified to alter the tumor microenvironment in favor of tumor immunity generally show that high-dose RT has a greater capacity to harness host anticancer immune defenses than low-dose RT [16]. Patients with CRPC have a high incidence of distant metastasis, and different therapies added to ADT have been studied to prolong survival. In the present study, we further examined the control of metastatic CRPC mediated by the abscopal effect of RT in mice treated with ADT. Hypofractionated regimens are reported to be better at inducing the proimmunogenic effects of radiation than single high/ablative doses [16,36]. Therefore, an RT regimen of 3 × 8 Gy was used to evaluate the abscopal effect in the present study. Increased cell death associated with augmented TILs (CD3+ T cells) was observed in secondary non-irradiated tumors of mice that received local RT compared to those that received sham RT.

IL-6 is capable of modulating diverse cell functions such as inflammatory reactions, and is implicated in the regulation of tumor growth and metastatic spread in different cancers [37]. IL-6 positively affects tumor development and is recognized as a key regulator of immunosuppression in advanced cancer [38]. An anti-IL-6 monoclonal antibody has been applied to treat metastatic CRPC in clinical studies [39]. We previously reported that IL-6 is crucial for aggressive tumor growth, MDSC recruitment, and a poor radiation response [5]. The data in the present study revealed that inhibiting IL-6 significantly decreased the size of secondary non-irradiated CPRC tumors, associated with augmented TILs and attenuated MDSC recruitment. Therefore, we suggest that in addition to a direct action on malignant cells, immune modulation mediated the augmented abscopal effect induced by anti-IL-6.

## 5. Conclusions

We suggest that RT increases the treatment response of metastatic CRPC to ADT through enhanced antitumor immunity in prostate cancer. Moreover, targeting IL-6 signaling could be a promising strategy for sensitizing prostate cancer to ADT + local RT in the clinic, particularly for metastatic CRPC.

## Figures and Tables

**Figure 1 cancers-11-00020-f001:**
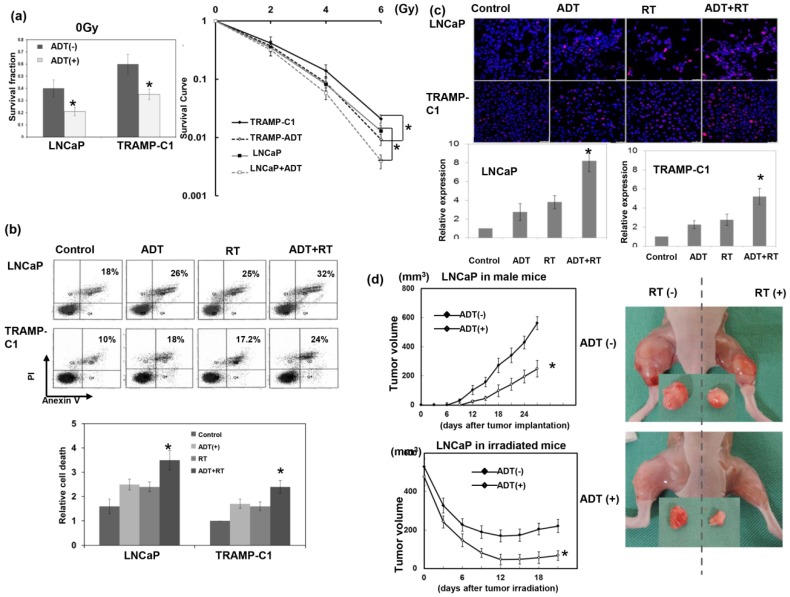
Effect of androgen deprivation therapy (ADT) on radiation sensitivities of prostate cancer. (**a**) Survival fractions were constructed using colony-forming assays for cells with or without ADT (10 μM enzalutamide). In addition, a clonogenic assay was performed with prostate cancer cells with or without 10 μM enzalutamide for 3 days before irradiation. The cells were irradiated with 0, 2, 4, or 6 Gy, and the survival curve was determined by colony counting and normalized with plating efficiency. Each point is an average of three experiments. (**b**) The in vitro effects of treatments on apoptosis as evaluated by fluorescence-activated cell sorting (FACS) with Annexin V-PI staining 48 h after irradiation. The *y* axis represents the ratio normalized by the value of TRAMP-C1 under control conditions. (**c**) DNA damage as evaluated by immunofluorescence staining with p-H2AX 24 h after irradiation. Scale bars: 50 μm. The quantification was the calculation of the value of the cell numbers positive for p-H2AX divided by the total cell number. The *y* axis represents the ratio normalized by the value under control conditions. (**d**) The effects of ADT on tumor growth curves and the radiation response were examined using LNCaP ectopic tumors in mice with or without medical ADT. We also showed the representative images 12 days after 15 Gy irradiation or sham irradiation (ADT, 25 mg/kg enzalutamide daily since one week before irradiation). The data represent the means of experiments (three animals in one experiment, performed three times independently), * *p* < 0.05.

**Figure 2 cancers-11-00020-f002:**
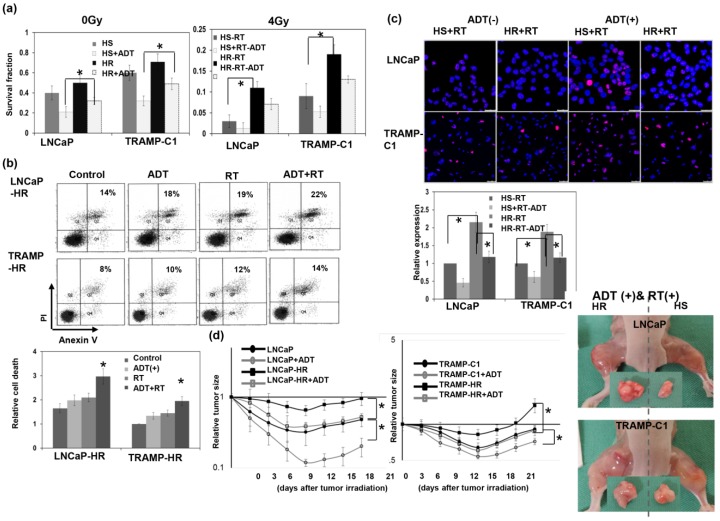
Treatment sensitivities of hormone-sensitive and hormone-resistant (HR) prostate cancer. (**a**) Prostate cancer (hormone-sensitive (HS) and the corresponding castration-resistant prostate cancer (CRPC)) cells were treated with ADT (10 μM enzalutamide for 3 days before irradiation) and irradiation with 0 or 4 Gy, and survival fractions were constructed using colony-forming assay data 10 days after treatment. (**b**) The in vitro effects of treatment-induced apoptosis as evaluated by FACS with Annexin V-PI staining. The *y* axis represents the ratio normalized by the value of TRAMP-HR under control conditions. (**c**) DNA damage as evaluated by immunofluorescence staining with p-H2AX. The *y* axis represents the ratio normalized by the value of HS cells with irradiation. Scale bars: 50 μm. (**d**) The effect of ADT on the radiosensitivity of prostate cancer (LNCaP, TRAMP-C1, and the corresponding CRPC cells), as demonstrated by tumor growth delay of the ectopic tumor after 15 Gy irradiation, and representative images 12 days after 15 Gy irradiation with or without ADT. The *y* axis shows the tumor volume ratio at each time point, divided by the tumor volume at irradiation. Data represent the means of experiments (three animals in one experiment, performed three times independently), * *p* < 0.05.

**Figure 3 cancers-11-00020-f003:**
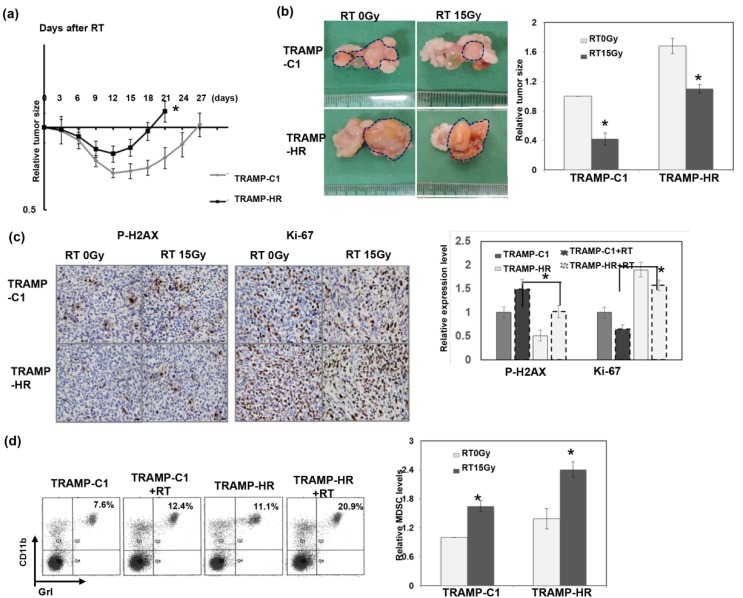
Radiation response of prostate cancer in an immunocompetent host. The radiosensitivity of prostate cancer (TRAMP-C1 and the corresponding CRPC cells), as demonstrated by (**a**) tumor growth delay of ectopic tumors after 15 Gy irradiation, and by (**b**) the orthotopic tumor model in immunocompetent mice. Representative images and quantitative data are shown 12 days after 15 Gy radiotherapy (RT) or sham irradiation. The *y* axis represents the relative ratio, normalized to the tumor size of TRAMP-C1 tumors in sham-irradiated mice (* *p* < 0.05). (**c**) Expression levels of Ki-67 and p-H2AX in tumors were evaluated by immunohistochemical staining (IHC) analysis. Scale bars: 20 μm. The *y* axis represents the ratio normalized by the value of TRAMP-C1 under control conditions. The recruitment of myeloid-derived suppressor cells (MDSCs) was evaluated by FACS using GrI-CD11b staining in the spleen (**d**). The MDSC level was the value of the number of CD11b+Gr1+ cells divided by the total cell number. The level of MDSCs in the tumors was evaluated by FACS **(e)** and by immunofluorescence using CD11b staining (**f**). Scale bars: 50 μm. The levels of interleukin (IL)-6 in the tumors were examined by immunofluorescence (IF) analysis (and in serum by ELISA with mice bearing tumors 48 h after 15 Gy RT or sham irradiation). (**g**). Columns: Means of three separate experiments; bars: SD; * *p* < 0.05.

**Figure 4 cancers-11-00020-f004:**
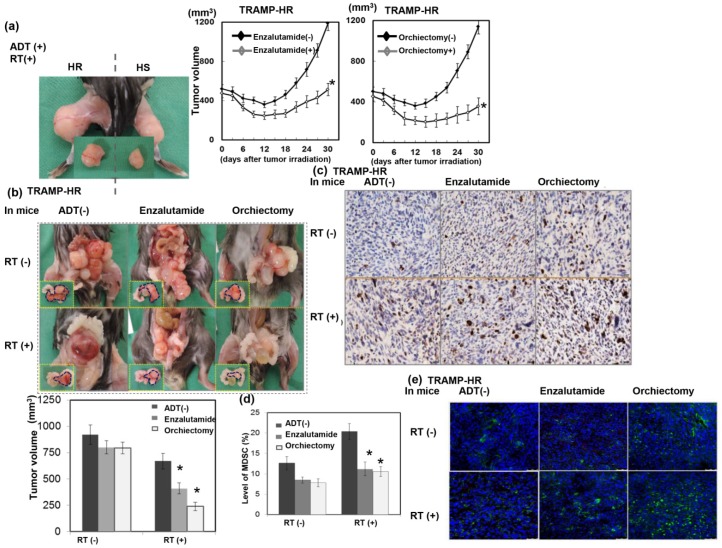
Role of ADT in the radiation sensitivity of CRPC in immunocompetent mice. The role of ADT on the radiosensitivity of CRPC was demonstrated by (**a**) tumor growth delay of TRAMP-HR ectopic tumors after 15 Gy irradiation. We also show the representative images 12 days after 15 Gy irradiation with or without ADT and by (**b**) the orthotopic tumor model 12 days after 15 Gy irradiation in immunocompetent mice. Representative images and quantitative data are shown. The *y* axis represents the relative ratio, normalized to the tumor size of TRAMP-HR tumors in sham-irradiated male mice (* *p* < 0.05). (**c**) Expression levels of p-H2AX were evaluated by IHC in CRPC tumors with different types of treatment. Scale bars: 20 μm. (**d**) The recruitment of MDSCs in the spleen evaluated by FACS using GrI-CD11b staining. The quantification for MDSC levels was the calculation of the value of the number of CD11b+Gr1+ cells divided by the total cell number. Columns: Means of three separate experiments; bars: SD; * *p* < 0.05. (**e**) The extent of tumor-infiltrating T cells (TILs), shown by immunofluorescence analysis, of tumor specimens using CD3 staining in CRPC tumors with different types of treatment. Scale bars: 50 μm

**Figure 5 cancers-11-00020-f005:**
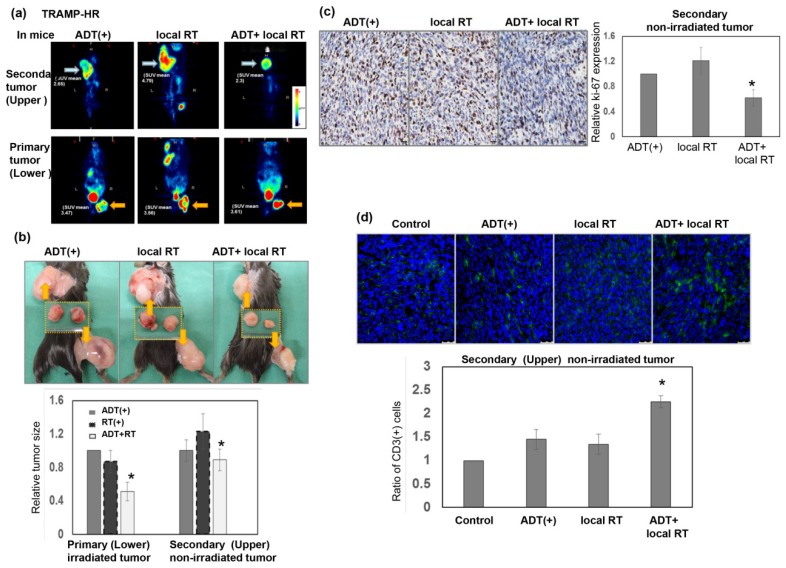
Role of RT in the response of metastatic CRPC to ADT. (**a**) Representative PET images from the tumor-bearing mice 24 h after three times 8 Gy to the right thigh tumor only (orange arrow: Primary irradiated tumor; blue arrow: Secondary non-irradiated tumor). (**b**) Representative images from the tumor-bearing mice 7 days after three times 8 Gy to the right thigh tumor only (right thigh tumor: Primary irradiated tumor; upper back tumor: Secondary non-irradiated tumor). The *y* axis represents the relative ratio, normalized to the tumor size of primary CRPC tumors in sham-irradiated mice treated with ADT (* *p* < 0.05). (**c**) Expression levels of Ki-67 of the secondary non-irradiated tumors were evaluated by IHC analysis. Representative images (Scale bars: 20 μm) and quantitative data are shown. The *y* axis represents the relative ratio, normalized to the value of secondary non-irradiated tumors in enzalutamide-treated mice (* *p* < 0.05). (**d**) The extent of TILs of the secondary non-irradiated tumors was examined by immunofluorescence analysis using tumor specimens that underwent CD3 staining in mice bearing ectopic tumors. Scale bars: 50 μm. The quantification was the calculation of the value of the cell numbers positive for CD3 divided by the total cell number. The *y* axis represents the ratio normalized by the value under control conditions.

**Figure 6 cancers-11-00020-f006:**
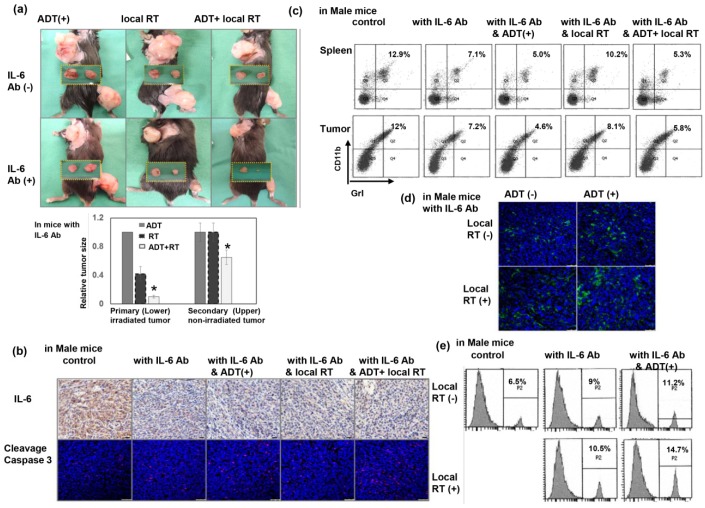
Inhibition of IL-6 enhanced the abscopal effect on metastatic CRPC. (**a**) The size of tumors (primary irradiated tumors and secondary non-irradiated tumors) was examined in CRPC-bearing mice with or without IL-6 blocking. Representative images and quantitative data are shown from the CRPC-bearing mice 7 days after three times 8 Gy to the right thigh tumor only (right thigh tumor: Primary irradiated tumor; upper beck tumor: Secondary non-irradiated tumor). The *y* axis represents the relative ratio, normalized to the tumor size of primary CRPC tumors in sham-irradiated mice treated with ADT and IL-6 antibodies (* *p* < 0.05). (**b**) Expression levels of IL-6 and cleavage caspase 3 of secondary non-irradiated tumors were evaluated by IHC (Scale bars: 20 μm) and IF (Scale bars: 50 μm) analysis, respectively. (**c**) Effect of IL-6 on the recruitment of MDSCs in the spleen and secondary non-irradiated tumors evaluated by FACS using GrI-CD11b staining in mice bearing CRPC tumors with or without IL-6 inhibition. The results are shown by representative slides. The effect of IL-6 on the number of TILs in secondary non-irradiated tumors was examined by immunofluorescence analysis using CD3 staining (**d**), and FACS using CD8 staining (**e**) in mice bearing CRPC with or without IL-6 inhibition. The results are shown by representative slides. Scale bars: 50 μm.

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
