# Peer review of "The Response of Prostate Cancer to Androgen Deprivation and Irradiation Due to Immune Modulation"

_cancers, 2018, doi:10.3390/cancers11010020_

Reviewer 1 Report

In this manuscript, Wu et al primarily performed studies using ectopic and othortopic tumor models in the mice, which could provide some clinical insights. They explored the anti-tumor effect of androgen deprivation therapy (ADT) and radiotherapy (RT) in vitro and in vivo, and the results are no surprise. Indeed, it has been well documented that ADT combined with RT could be particularly beneficial for prostate cancer (PCa) patients with high-risk disease. Recent RADAR clinical trial compared two different lengths of ADT treatment combined with RT, and showed 18 months of ADT may be optimal for locally advanced PCa. Using cellines that are directly derived from transgenic mice (TRAMP), they demonstrate that CRPC cells harbor higher IL-6 level. They also investigated the immunological actions that could account for the abscopal effect. The authors conclude that the immunosuppressive microenvironment, which is a feature of advanced PCa should be exploited for developing better therapies.

A few suggestions for this manuscript:

1)Figure1:  The graphs in Figure 1a should be combined as shown in Fig 1b and c.

There is no information on clonogenic assay.

2)Figure2: Quantified data should be shown for Figure 2b and c. It is difficult to differentiate the curves representing individual conditions. In Fig 2a, the conditions that show significant difference actually are already separated when there is 0 Gy, and increasing dose of irradiation doesn’t seem to separate the curves more (run parallel).

3)Figure3: The orthotopic tumor area should be outlined in 3b (same in 4b) in order to be better appreciated.  The interpretation for fig3 should be that HR tumors are more resistant to the IR than HS. In c, it should not be “had less RT-induced DNA damage” (HR almost had two-fold increase), instead the data shows that even with increased DNA damage HR is still lower than HS.

4)Figure5: It is an overstatement to use the abscopal effect to relate to the metastatic CRPC which happens in organs like bone, lung etc (same as fig6).

5)Figure6: Conditions without IL-6 treatment should be included in order to conclude the enhanced effect of IL-6. 

6)Did authors look into the invasion in the orthotopic model as mentioned in the methods?

Author Response

Response to Reviewer #1:   Thanks the reviewer’s comments.The manuscript has undergone English language editing by MDPI. In addition, the changes that are in response to the Reviewer’s comments are shown in text with grey highlight and for reorganization are shown in blue text in the revised manuscript ( please see attched pdf.file)

Reviewer 2 Report

The paper entitled: "The response of prostate cancer to androgen deprivation and irradiation related to immune modulation" is an interesting paper describing the additive effect of ADT and RT in reducing tumour growth, effect which is reduced if cells have first been rendered resistant to androgen. Moreover, the authors demonstrate the involvement of TIL in this response.

However, before the paper can be published the authors need to change the following:

- Why did the author chose LNCaP which androgen-sensitive but not dependent therefore naturally grow without any androgen, for their model? It would have been better to use a cell line like PC346 which is androgen dependant and its derivative PC346Flu1 and PC346Flu2 which have already been rendered androgen independent for their study.

- Where did the author purchased their TRAMP-C1, it is not written in the method section.

- Experiments relating to tumour implantation are not clear. Did the authors really use 1x10^6 cells when the cells when injected subcutaneously but 6x10^6 when the cells where injected directly into the prostate? What volume did they use? What does "intraoperatively" mean?

- How did the authors measured tumour invasion?

- Supplementary section is not available.

- The method for the TIL extraction is not mention, reference 18 quoted is a review!!!

- Line 110 why do the authors write: "Furthermore, we used an antibody specific to CD3 to define tumour-infiltrating T cells (TILs) when they have already written in line 108 that they have used a CD3 antibody at 1:20 dilution?

- The results mentioned several times colony forming assay but no method has been provided.

- How did the authors calculate their "survival fraction", all their "relative expression", "relative tumour size" etc...

- Flow cytometry should be revised as the do not look sufficiently compensated.

- How was the number of positive cells assessed/coutned in all the flurorescence experiments? was is only one person?

Author Response

Response to Reviewer #2: 

Thanks the reviewer’s comments. According to the reviewer’s suggestion, the manuscript has undergone English language editing by MDPI. In addition, the changes that are in response to the Reviewers comments are shown in text with grey highlight and for reorganization are shown in blue text in the revised manuscript (Please see the attached pdf. file)

Round  2

Reviewer 1 Report

After reviewing the revised manuscript, I think the authors have addressed all my comments and the manuscript is improved to warrant publication in Cancers.

Best,